# A General and Efficient Approach for the Dual-Scale Infiltration Flow Balancing in In Situ Injection Molding of Continuous Fiber Reinforced Thermoplastic Composites

**DOI:** 10.3390/polym13162689

**Published:** 2021-08-12

**Authors:** Xiansong He, Yi Liu, Wangqing Wu

**Affiliations:** 1State Key Laboratory of High Performance Complex Manufacturing, Central South University, Lushan South Road 932, Changsha 410083, China; csuxshe@csu.edu.cn (X.H.); csuhxs2021@163.com (Y.L.); 2School of Mechanical and Electrical Engineering, Central South University, Lushan South Road 932, Changsha 410083, China

**Keywords:** in situ injection molding, continuous fiber reinforced thermoplastic composites, dual-scale infiltration flow, capillary effect, capillary number

## Abstract

In situ injection molding of continuous fiber reinforced thermoplastic composites is challenged by unbalanced dual-scale infiltration flow due to the pronounced capillary effect. In this paper, a general and efficient approach was proposed for dual-scale infiltration flow balancing based on numerical simulation. Specifically, Stokes and Brinkman equations were used to describe the infiltration flow in inter- and intra-fiber bundles. In particular, capillary pressure drop was integrated in the Brinkmann equation to consider the capillary effect. The infiltration flow front is tracked by the level set method. Numerical simulation and experimental results indicate that the numerical model can accurately demonstrate the unbalanced infiltration flow in inter- and intra-fiber bundles caused by the changes of the injection rate, the resin viscosity, the injection rate, the fiber volume fraction and the capillary number. In addition, the infiltration flow velocity in inter- and intra-fiber bundles can be efficiently tuned by the capillary number, which is mainly determined by the injection rate for a specified resin system. The optimal capillary numbers obtained by simulation and experiment are 0.022 and 0.026, which are very close to each other. Finally, one-dimensional in situ injection molding experiments with constant injection pressure were conducted to prepare fiber reinforced polymerized cyclic butylene terephthalate composite laminate with various flow rates along the infiltration direction. The experimental results confirmed that the lowest porosity and the highest interlaminar shear strength of the composite can only be obtained with the optimized capillary number, which is basically consistent with the simulation results.

## 1. Introduction

Continuous fiber reinforced thermoplastics (CFRTs) are sustainable composite materials [1,2,3,4], which have great potential in automobiles [5], face guards [6] and other industrial fields where lightweight [7], environmentally-friendly [8] and good mechanical properties are needed. In recent years, the preparation of CFRTs has also attracted the close attention of scholars [9,10,11]. Generally, CFRTs are mainly prepared by the melting method [12] and reaction molding [13]. Panthapulakkal et al. found that wheat straw fibers can be used as potential reinforcing materials for making thermoplastic composites [14]. Saiter, et al. prepared bamboo fiber reinforced thermoplastic matrix composites by extrusion and injection [15]. Rijswijk et al. optimized the resin formula of GF/PA6 composites formed by vacuum assisted resin infusion (VARI) and studied the influence of polymerization temperature on the mechanical properties of the composites [16]. Kouan, et al. prepared WBF/pCBT composites by a combination of the solvent method and molding process [17]. Yan et al. prepared continuous glass fiber reinforced anionic polyamide-6 composites via in situ ring-opening polymerization of caprolactam monomers [18]. Up to now, the molding processes has mainly focused on the solvent method, extrusion and injection, compression molding, VARI and other processes. However, the in situ injection molding process [19] with high productivity and easy automation is rarely studied. Unlike the melting method, in in situ injection molding, low viscosity monomer or oligomer is impregnated with reinforcing fiber to form thermoplastic composite material after ring-opening polymerization in the mold. In situ injection molding technology can significantly reduce the manufacturing cost of high-performance fiber reinforced thermoplastic resin matrix composites. It has great market potential in realizing large-scale and high-level manufacturing of thermoplastic composite materials.

Presently, one of the main challenges of in situ injection molding is to control the flow balance between and within fiber bundles during the infiltration of fiber cloth preforms [20]. Generally, the fiber cloth structure of CFRTs is a dual-scale structure. Fiber cloth consists of fiber bundles arranged in a regular way, and fiber bundles consist of hundreds or thousands of fiber filaments. For conventional thermoplastic resins, because of the high melt viscosity, the injection pressure in the molding process is much higher than the capillary pressure [21,22], so the influence of capillary pressure can be ignored. However, due to the extremely low viscosity of the low molecular thermoplastic resin system [23,24], the pressure required for the infiltration is also very small (less than 1 MPa). The capillary effect of resin is pronounced during fiber cloth infiltration, which is affected not only by injection pressure but also by capillary pressure [25,26,27]. Capillary pressure can hinder or promote the osmotic flow in fiber bundles, resulting in the formation of pores in or between fiber bundles. The porosity in the composite material will seriously affect the mechanical properties of the parts, such as bending strength and fatigue strength [28,29,30,31,32]. Therefore, it is necessary to further study the influencing mechanism of capillary pressure on dual-scale infiltration so that the balance can be achieved through the flow front, and the porosity of the molded part can be reduced to improve the mechanical properties.

Although this kind of dual-scale infiltration problem has been widely studied in liquid composite molding [33,34], it still has considerable challenges in in situ injection molding. To solve this problem, Verry et al., based on Darcy’s Law, proposed a method to measure capillary pressure of multiaxial warp knitted fabric (NCF) with low molecular reactivity and low viscosity resin penetration [26]. It was proved that capillary pressure has a significant influence on infiltration. Zingraff et al. measured the capillary pressure drop of polyamide 12 resin and determined its optimum capillary number [35]. Leclerc et al. used RTM equipment to carry out unidirectional infiltration experiments under constant pressure and calculated the porosity of parts at different positions through experiments [36]. Lawrence et al. developed a set of equipment to study the double-scale effect during resin infiltration [37]. Francois et al. established the mathematical model of capillary ascending flow front of the fiber bundle [38]. Through experimental verification, it was found that when the capillary number is between 1 × 10^−3^ and 1 × 10^−6^, the capillary pressure plays a leading role, and the inner flow of fiber bundles is faster. Härter et al. used PAM-RTM software to simulate the flow of resin in reinforcing medium and made a numerical study on the influence of transverse permeability of resin flow [39]. Maung et al. studied the influence of fabric weaving mode on the duration of the vacuum impregnation process in the PAM-RTM program and simulated the porosity of different materials [40].

In summary, researchers have done extensive experimental research on the dual-scale infiltration process. That research mainly focused on injection technology and macroscopic observation of experiments. In terms of numerical modeling and simulation, a dedicated commercial finite element software, PAM-RTM, has been developed, especially for the liquid molding of structural thermosetting composite materials. When it comes to the in situ injection molding of CFRTs, however, there have been no general and effective tools for the simulation of dual-scale infiltration. In this work, an effective method is proposed to simulate dual-scale infiltration based on a general commercial finite element software, COMSOL Multiphysics. Stokes and Brinkman equations were used to mathematically model the infiltration flow between fiber bundles and within fiber bundles, respectively. The level set method was used to track the flow front. The numerical model of dual-scale infiltration was established in COMSOL. Dual-scale infiltration flow balancing in in situ injection molding was studied by means of experiment and simulation. The influence of injection rate, resin viscosity, fiber volume fraction and capillary number on the dual-scale osmotic flow balance was investigated. Textile fiber reinforced polymerized cyclic butylene terephthalate (pCBT) matrix composite laminates with different capillary numbers were prepared by an in-house developed in situ injection molding device. The composite samples were subjected to the three-point bending test and optical microscope observation to characterize their interlaminar shear strength and pore distribution, respectively.

## 2. Materials and Methods

### 2.1. Formation Mechanism of Dual-Scale Infiltration Flow

It has been confirmed that there are dual-scale porous structures in various liquid composite molding technologies including in situ injection molding [41,42,43,44]. As shown in Figure 1, the inherent structural characteristics further lead to the phenomenon of dual-scale infiltration flow. In addition to injection pressure, the resin units in the fiber bundle are also subjected to capillary pressure. Generally, capillary pressure is almost negligible because it is much smaller than injection pressure. However, when the capillary effect becomes obvious due to extremely low resin viscosity, it cannot be ignored any longer. When the capillary pressure promotes resin infiltration, the flow inside fiber bundles is faster than the flow between fiber bundles, thus forming macroscopic pores between fiber bundles. When capillary pressure hinders resin infiltration, the flow in the fiber bundles is slower than that between fiber bundles, thus forming microscopic pores in the fiber bundles. Patel et al. found that there is an optimal infiltration speed so that the flow in inter- and intra-fiber bundles can be balanced to minimize pore formation during infiltration [45].

### 2.2. Mechanism of Capillary Pressure

The schematic diagram of infiltration is shown in Figure 2. In the process of in situ injection molding resin infiltrated fiber cloth, the interface between fiber and resin gradually replaces the interface between air and fiber. Due to the difference of surface tension between resin and air, a pressure gradient is generated at the flow front of resin. This effect has been widely studied in soil science and hydrology and has been gradually used in the study of composite molding processes [46]. Assuming that the infiltration process is saturated flow, capillary pressure can be defined as the work that resin needs to do to replace air in a unit volume of porous media. When the capillary pressure is negative, this means that the resin will automatically infiltrate the fiber without external force. When the capillary pressure is positive, it is necessary to overcome the capillary pressure to infiltrate the fiber. Based on thermodynamic theory [47], capillary pressure can be calculated by Equation (1).
(1)$$\Delta P_{\gamma }=-S_{f}\gamma_{\mathrm{ma}}\cos(\theta )$$
where $\;\Delta P_{\gamma }$ is the capillary pressure; $S_{f}$ is the interface area between fiber and resin in unit resin matrix; $\gamma_{\mathrm{ma}}$ is the surface tension between resin and air; *θ* is the dynamic contact angle.

Moreover, many studies show that the balance between capillary pressure and flow front in the process of dual-scale infiltration in in situ injection molding is largely affected by the dimensionless parameter capillary number (Ca) [48,49]. The capillary number can be calculated by Equation (2).
(2)$$Ca=\frac{u\mu }{\gamma_{\mathrm{ma}}}$$
where u is infiltration velocity; μ is the resin viscosity.

### 2.3. Numerical Simulation

In this paper, the physical model and boundary conditions are established based on the actual experimental process. The numerical simulation model of dual-scale infiltration was established by using COMSOL [50,51,52,53]. Based on the mechanism of formation of dual-scale infiltration flow and capillary pressure, the effects of injection rate, resin viscosity, fiber volume fraction and capillary number on the dual-scale infiltration flow were studied.

#### 2.3.1. Physical Model

Based on the cross-sectional morphology of composite material formed by fiber cloth and resin system shown in Figure 3a, a physical model of fiber reinforcement dual-scale structure was established, as shown in Figure 3b. The dimensions shown in the model were based on the actual dimensions of the fiber cloth used in the in situ injection molding experiment. According to the data provided by the manufacturer, the porosity of the fiber bundle is 48%, and the density of resin is 1150 kg/m^3^. The permeability is $2.2\times 10^{-10}$ $m^{2}$, calculated by the Gebart formula.

#### 2.3.2. Mathematical Modeling

##### Control Equation between Fiber Bundles:

In this study, the channel size between bundles of fiber cloth was less than 100 um, so the Stokes flow equation could be used for numerical simulation [54,55]. The Stokes flow equation is mainly applied to micro-fluid flow with extremely small geometry size, which is another manifestation of the Navier–Stokes equation [56,57,58], ignoring the inertia term. Assuming that the fluid is incompressible, the Stokes flow equation can be described as momentum equation (Equation (3)) and continuity equation (Equation (4)):(3)$$\rho \cdot \frac{\partial u}{\partial t}=-\nabla p+\nabla \cdot \mu \left(\nabla u+\nabla u^{T}\right)$$
(4)ρ⋅∇u=0
where u is the fluid flow velocity; *ρ* is fluid density; p is pressure.

##### Control Equation in Fiber Bundles:

The Brinkman equation [59] not only considers the viscous shear stress term of fluid in the Navier–Stokes equation, but also considers Darcy’s law [60]. This equation can well explain the rapid flow of fluid in porous media. Because the mold temperature is equal to the resin temperature during infiltration, the law of conservation of energy is not considered. Assuming that fluid is incompressible and substituting the force term with the capillary pressure gradient, the Brinkman equation can be described as the momentum equation (Equation (5)) and continuity equation (Equation (6)).
(5)$$\frac{\rho }{\varepsilon_{p}}\cdot \frac{\partial u}{\partial t}=-\nabla p+\nabla \cdot \frac{\eta }{\varepsilon_{p}}\left(\nabla u+\nabla u^{T}\right)-\frac{\eta }{K}\cdot u+\frac{\Delta P_{\gamma }}{L}$$
(6)ρ⋅∇u=0
where $\varepsilon_{p}$ is porosity; *K* is the permeability of porous media; $\Delta P_{\gamma }$ is capillary pressure; *L* is the infiltration length of fiber bundle.

#### 2.3.3. Level Set Tracking

The level set method is a technique that uses a fixed grid to represent a moving interface or boundary [61,62]. This simulation used the level set method to track the interface of the resin flow front. Firstly, the level set variables were initialized, so that the level set variables changed smoothly from 0 to 1 at the interface of the resin flow front. In the process of numerical simulation of the level set tracing flow front, phase initialization and transient analysis are greatly needed. Phase initialization is conducted to calculate the distance from the entrance to the initial interface. Then transient analysis uses different initialization conditions according to the divided areas. In the area outside the interface, that is, when ϕ> 0.5, the initial value of the level set variable of Equation (7) is adopted. In the area inside the interface, that is, when ϕ< 0.5, the initial value of the level set variable of Equation (8) is adopted.
(7)$$\phi_{0}=\frac{1}{1+e^{-\frac{D}{\varepsilon }}}$$
(8)$$\phi_{0}=\frac{1}{1+e^{\frac{D}{\varepsilon }}}$$
where *D* is the distance from the entrance to the initial interface.

#### 2.3.4. Boundary Conditions

The simulation took the actual parameters of the experiment as boundary conditions. Under the condition of constant flow velocity, the boundary conditions at the entrance were set as the normal inflow velocity during the experiment. The pressure boundary was set at the exit of the fiber bundle area. The pressure was the capillary pressure measured under certain experimental conditions, while the outlet pressure of the channel between fiber bundles was 0. The outer wall surface of the physical model was set to have no slip. The specific modeling situation was as follows:

Initial conditions: *p* = 0, *u* = 0.

Boundary conditions of channel flow between fiber bundles: *u_inj_* = injection rate, *P_out_* = 0.

Flow boundary conditions in fiber bundles: *u_inj_* = injection rate, *P_out_* = Δ*P*_γ_.

### 2.4. Experimentation

#### 2.4.1. Constant Flow Rate Infiltration Flow Characterization

Based on the theoretical analysis of dual-scale infiltration flow, it is clear that injection rate, resin viscosity, fiber volume fraction and capillary number have an influence on dual-scale infiltration. The self-built experimental platform was used to test the capillary pressure and flow front changes under different infiltration conditions. The experimental results will be combined with simulation research to guide the preparation of thermoplastic pCBT-based composites.

##### Materials

Because cyclic butylene terephthalate (CBT) resin will solidify at low temperature, and the melting temperature of CBT resin is 200 °C, during the experiment, the cleaning process was very time consuming. Moreover, it can be seen from Figure 4a,b that the viscosity and surface tension of the CBT resin system and epoxy resin system showed a high similarity in the research range. Therefore, the dual-scale infiltration experiment replaced CBT resin by epoxy resin material; when different resin infiltrates the same fiber volume fraction, the optimal capillary number will not change. The rule of the combined research can be obtained to guide the dual-scale infiltration of CBT resin system. The resin was produced by Momentive Specialty Chemicals, Germany, the specification was RIMR135, and the density was 1.15 g/cm^3^. The fiber cloth was uniaxial glass fiber cloth with an areal density of 0.085 g/cm^3^ and a density of glass fiber of 2.5 g/cm^3^.

##### Experimental Platform

The structure system of the dual-scale infiltration experiment platform is shown in Figure 5. Specifically, the thermostatic bath was used to control the temperature of resin in the beaker. An injection pump was used to inject the resin at constant temperature into the mold cavity to wet the fiber cloth. The conveying pipeline and its thermal insulation device were responsible for conveying the resin and maintaining the temperature during the conveying process so as to ensure that the viscosity performance of the resin remained constant. A micro flow meter was used to measure resin flow. A three-way valve was used to control the flow direction of resin in the experimental process and control the experimental process. The pressure sensor was responsible for collecting the pressure change curve at the entrance of the mold used to obtain the capillary pressure in infiltration. The infiltration mold provided a closed cavity for the infiltration experiment and controlled the internal temperature of the mold. The glass above the mold was convenient to observe the position and shape of the flow front. The mold cavity size was 220 mm × 140 mm × 2 mm. The image acquisition device was used to record the position and shape of the resin flow front. The resin collection tank was used to collect the excess resin flowing out of the mold outlet. The main technical parameters of the experimental platform are shown in Table 1.

##### Experimental Principle

Under the condition of constant flow injection, the relationship between the difference between mold inlet pressure and atmospheric pressure and time was recorded. Because the pressure sensor was zeroed during the experiment, the change of the inlet pressure actually measured was the difference between the inlet pressure and the atmospheric pressure. According to Darcy’s law, under the condition of constant flow rate, the theoretical equation is shown as Equation (9), and the capillary pressure $\Delta P_{\gamma }$ is the intercept of the equation on the ordinate axis.
(9)$$P_{\mathrm{inj}}-P_{\mathrm{atm}}=\frac{Q^{2}\eta }{2A^{2}K\left(1-V_{f}\right)}t+∆P_{\gamma }$$
where $P_{\mathrm{inj}}$ is the inlet pressure; $P_{atm}$ is atmospheric pressure; *Q* is the injection flow rate; *η* is resin viscosity; *A* is the cross-sectional area of the mold cavity perpendicular to the infiltration direction; *K* is permeability; $V_{f}$ is the fiber volume fraction.

Moreover, the injection rate was controlled by the volumetric flow rate. With the change of injection rate, the capillary number changed correspondingly. The temperature of epoxy resin was controlled by controlling the temperature of the constant temperature and humidity tank, so as to control the viscosity of epoxy resin. The fiber volume fraction was kept constant by controlling the number of layers of fiber cloth. During the experiment, the capillary pressure in the infiltration process was calculated by the pressure sensor data. The position and flow balance state of the flow front in the infiltration process were obtained using a high-definition camera.

#### 2.4.2. Composites Preparation and Characterization

According to the conclusions of the simulation and experiment of dual-scale infiltration flow of fiber bundles, continuous glass fiber reinforced thermoplastic pCBT matrix composites were prepared by self-developed in situ injection molding device. The correctness of the experimental and simulation conclusions was further verified by observing the distribution of cross-section pores and measuring the mechanical properties of composite parts.

##### In Situ Injection Molding Device

To realize the research of the batch controllable manufacturing and molding process of thermoplastic composites, an in situ polymerization injection molding device integrating resin melting and conveying on-line mixing and composite molding was developed. The existing in situ polymerization injection molding equipment has the disadvantages of low automation degree, long processing cycle, etc. To solve these problems, this paper provided an integrated device for melt conveying and on-line mixing with high automation and high efficiency, and its main technical parameters are shown in Table 2. Specifically, Figure 6a shows the schematic diagram of the in situ polymerization injection molding device. The resin enters the resin continuous melting and conveying module from the hopper for melting and conveying. The melted resin enters the high-precision metering module for metering and is delivered to the cross. The catalyst enters the catalyst continuous conveying module for conveying. Then, after being metered by the high-precision metering module, it merges with the resin at the four-way outlet and enters the continuous on-line mixing module for mixing. In the whole process, the heating and insulation of the device is completed by the temperature control module. A physical drawing of the device is shown in Figure 6b.

##### Preparation and Characterization of Composite Materials

According to the results of the numerical simulation and experiment, the resin used in the experiment was low molecular thermoplastic CBT500 resin and catalyst system. CBT500 resin is a common low molecular thermoplastic CBT500 resin in industrial production. In the preparation stage of the reaction, the viscosity of the resin system was 35 mPa·s, and the surface tension was 34.5 mN/m. The fiber cloth was a uniaxial warp knitted fabric with a volume fraction of 46% and a permeability of 2.2 × 10^–10^. Moreover, the composite laminate was prepared under constant pressure injection mode so that various injection rates could be obtained in one specimen. The relationship between infiltration velocity and flow front position when the injection pressure was 20 kPa is shown in Figure 7a, and the average infiltration velocity was different in three different regions. At a distance of 10 mm to 30 mm from the entrance, the average infiltration velocity was 25 mm/s in region 1. At a distance of 40 mm to 60 mm from the entrance, the average infiltration velocity was 12 mm/s in region 2. At a distance of 100 mm to 120 mm from the entrance, the average infiltration velocity was 5 mm/s in region 3. The continuous glass fiber reinforced thermoplastic pCBT matrix composite prepared by the in situ polymerization injection molding device is shown in Figure 7b.

According to the infiltration velocity, the viscosity and surface tension of the resin of the three regions, the capillary numbers were calculated by Equation (2) as 0.007, 0.018 and 0.037, respectively. Therefore, the influence of different capillary numbers on pore distribution and mechanical properties of composite materials can be studied.

Then the pipeline temperature and mold temperature of the in situ polymerization injection molding control system were set at 190 °C. The CBT500 resin particles were put into a hopper for melting and plasticizing, mixed with DBTL catalyst through a mixing head and then injected into the mold cavity at constant pressure. After full infiltration, the in situ injection molding device and the inlet and outlet of the mold were closed, and the mold temperature was kept at 190 °C. After the mold temperature was maintained for 30 min, the mold was allowed to cool to room temperature, and then the product was taken out. Finally, the composite materials in three different regions were cut and sanded into rectangular plates with dimensions of 20 mm × 10 mm × 2 mm.

Pores of composite materials reduce the strength, stiffness and fatigue strength of composite materials. In this study, the pore distribution of composite materials was observed by an ultra-depth-of-field three-dimensional microscope. Specifically, the composite plate was cut into small pieces with the size of 10 mm × 10 mm × 2 mm according to different regions. Then the cut material was put into the metallographic sample mosaic machine, the mosaic powder was embedded, the heating temperature was set to 140 °C and the metallographic sample was taken out after the mosaic powder was solidified. Then, the metallographic sandpaper was placed on a polishing machine, and after the surface was smooth, it was polished with abrasive instead of soft sandpaper to prepare the metallographic fiber observation sample. Finally, the metallographic specimen was placed in a super depth-of-field three-dimensional microscope for pore distribution detection. The mechanical properties were tested by a CMT4204 universal mechanical testing machine using the ISO13140 testing standard. The size of the test piece was 20 mm × 10 mm × 2 mm, the span was 10 mm and the loading speed was 1 mm ± 0.2 mm/min. Calculation of interlaminar shear strength (ILSS) is shown in Equation (10).
(10)$$\tau =\frac{3}{4}\times \frac{F}{bh}$$
where *F* is the maximum load; *b* is the width of the sample; *h* is the thickness of the experiment.

## 3. Results and Discussion

### 3.1. Simulation Result Analysis

#### 3.1.1. Influence of Injection Rate on Infiltration Flow

In this study, the process parameters were set as mold temperature of 120 °C, resin viscosity of 30 mPa·s, and injection rates of 24, 36, 42 and 48 mm/s, respectively. Figure 8a shows the influence of injection rate on the flow velocity in the fiber bundle and the channel between bundles. With the increase of injection rate, the flow velocity in the intra- and inter-fiber bundle channels gradually increased. However, the increasing rate of the flow velocity in the inter-bundle channel was obviously greater than that in the fiber bundle. When the injection rate was 24 mm/s, the flow velocity in the fiber bundle was faster than that in the inter-bundle channel. When the injection rate was 36 mm/s, 42 mm/s and 48 mm/s, the flow velocity in the inter-bundle channel was faster than that in the fiber bundle. Figure 8b shows the influence of the injection rate on the flow front in the dual-scale infiltration. With the increase of injection rate, the moving velocity of the flow front was accelerated, and different injection rates led to different flow patterns at the flow front. When infiltration time t = 0.04 s, with the increase of injection rate and capillary pressure, the flow velocity in the fiber bundle was gradually slower than that in the inter-bundle channel. When t = 0.08 s, the velocity difference between the two regions became more and more obvious. When the resin flowed to the warp knitting line at the end, the resin would obviously wet transversely along the warp knitting line here, which would lead to the generation of pores at the end of the fiber bundle. With the increase of injection rate and velocity difference, the position of pores gradually concentrated from edge to fiber bundle.

#### 3.1.2. Influence of Resin Viscosity on Infiltration Flow

According to Figure 4a, there is a negative correlation between resin viscosity and resin temperature, so this study changed the resin viscosity by controlling the resin temperature. The resin viscosity was set at 226,162,125 and 30 mPa·s when the injection rate was 24 mm/s. Figure 9a shows the calculation result of the influence of viscosity on the flow velocity in the fiber bundle and the channel between bundles. With the increase of resin viscosity, the flow velocity in the fiber bundles decreased gradually, while the flow velocity in the channel between fiber bundles increased gradually. Figure 9b shows the influence of resin viscosity on the flow front in the dual-scale infiltration. When the infiltration time t = 0.05 s, with the increase of resin viscosity and capillary pressure, the flow velocity in the fiber bundle was gradually slower than that in the inter-bundle channel. When t = 0.15 s, the velocity difference between the two regions became more and more obvious. When t = 0.25 s, the resin had already flowed to the warp knitting line at the end, where the resin obviously wetted transversely along the warp knitting line, resulting in the following pores at the end of the fiber bundle. When the viscosity was 30 mPa·s, the flow velocity in the fiber bundle was faster than that in the inter-bundle channel, so pores would be generated in the inter-bundle channel. When the viscosity was 125, 162 and 226 mPa·s, the flow velocity in the fiber bundle was slower than that in the inter-bundle channel, so pores would be generated in the fiber bundle. With the increase of viscosity, the flow velocity difference would increase, and the position of pores would gradually concentrate from the edge to the inside of the fiber bundle.

#### 3.1.3. Influence of Fiber Volume Fraction on Infiltration Flow

In this study, according to the fiber volume fraction commonly used in the relevant literature [63,64,65] and the actual production process, the problem of material strength was also considered. Moreover, in situ injection molding was essentially similar to resin transfer molding (RTM), which can also be called thermal resin transfer molding. The fiber volume fraction of RTM molding composites is generally about 50%. Therefore, when the injection rate was 24 mm/s and the resin viscosity was 30 mPa·s, the fiber volume fraction was 46, 52 and 62%. Figure 10a shows the calculation result of the influence of fiber volume fraction on the flow velocity in the fiber bundle and the channel between bundles. With the increase of fiber volume fraction, the flow velocity in fiber bundles decreased gradually, while the flow velocity in the channel between fiber bundles increased gradually. When the fiber volume fraction was 52% and 62%, the flow velocity of resin between fiber bundles was faster than that inside fiber bundles. When the fiber volume fraction was 46%, the flow velocity in the fiber bundles was faster than that in the channels between the fiber bundles. Figure 10b shows the influence of the fiber volume fraction on the flow front in the dual-scale infiltration. With the increase of fiber volume fraction and capillary pressure, the flow velocity in the fiber bundle was gradually slower than that in the channel between the bundles. When the resin flowed to the warp knitting line at the end, obvious transverse infiltration would occur along the warp knitting line here, which would lead to pores at the end of the fiber bundle. Moreover, with the increase of fiber volume fraction and velocity difference, the position of pores gradually concentrated from the edge to the inside of fiber bundle, and the size of pores gradually increased.

#### 3.1.4. Influence of Capillary Number on Infiltration Flow

Capillary number is composed of resin viscosity, injection rate and surface tension, which is a dimensionless parameter for comprehensively evaluating the influence of infiltration parameters on the infiltration process. The capillary number was set to be 0.018, 0.028, 0.072, 0.091 and 0.127 respectively, and the influence of capillary number on flow velocity is shown in Figure 11a. When the capillary number was 0.127, 0.091, 0.072 and 0.028, the flow velocity in the channel between fiber bundles was faster, and air pockets were formed inside the fiber bundles, thus forming pores. When the capillary number was 0.018, it could be seen that the flow velocity in the fiber bundle was faster, and the optimal capillary number was about 0.022. Figure 11b shows the influence of capillary number on the flow front in the dual-scale infiltration. When the capillary number was 0.018, the flow velocity in the fiber bundle was faster than that in the inter-bundle channel, and the flow front in the fiber bundle protruded forward, and the pore at the end was generated in the inter-bundle channel. With the increase of capillary number, because of the increase of capillary pressure, the flow velocity in the fiber bundle was gradually slower than that in the channel between bundles. When the resin flowed to the end warp knitting line, the resin would obviously wet transversely along the warp knitting line here, which led to the generation of pores at the channel between fiber bundles. Moreover, with the increase of capillary number and velocity difference, the position of pores gradually concentrated from edge to fiber bundle, and the size of pores gradually increased.

### 3.2. Experimental Results Analysis

#### 3.2.1. Influence of Injection Rate on Capillary Pressure of Infiltration Flow

The influence of the injection rate on the inlet pressure when the temperature was 120 °C and the viscosity was 30 mPa·s is shown in Figure 12a. With the increase of injection rate, the maximum inlet pressure gradually increased. The maximum inlet pressure was about 15 kPa at 24 mm/s, 25 kPa at 36 mm/s, 35 kPa at 42 mm/s and 40 kPa at 48 mm/s. With the increase of injection rate, the intercept increased gradually, which indicated that the capillary pressure increased gradually during infiltration. In addition, the capillary pressure was about −6 kPa at 24 mm/s, 2 kPa at 36 mm/s, 4 kPa at 42 mm/s and 11 kPa at 48 mm/s. This also verified the changing trend of flow velocity with the increase of injection rate in simulation.

#### 3.2.2. Influence of Resin Viscosity on Capillary Pressure of Infiltration Flow

The influence of resin viscosity under different temperature conditions on the inlet pressure changes when the injection rate was 24 mm/s is shown in Figure 12b. With the increase of resin viscosity, the maximum inlet pressure gradually increased. The maximum inlet pressure was about 120 kPa at 226 mPa·s, 90 kPa at 162 mPa·s and 58 kPa at 125 mPa·s. With the increase of resin viscosity, the intercept increased gradually, which indicated that the capillary pressure increased gradually during infiltration. The resin had a capillary pressure of about 45 kPa at 226 mPa·s, about 30 kPa at 162 mPa·s and about 26 kPa at 125 mPa·s. The capillary pressure accounted for 37.5%, 33.3% and 44.8% of the maximum inlet pressure under three resin viscosity conditions, respectively.

#### 3.2.3. Influence of Fiber Volume Fraction on Capillary Pressure of Infiltration Flow

The influence of different fiber volume fractions on the inlet pressure change when the temperature was 120 °C, the viscosity was 30 mPa·s and the injection rate was 24 mm/s, is shown in Figure 12c. With the increase of fiber volume fraction, the maximum inlet pressure gradually increased. With the increase of fiber volume fraction, the intercept gradually increased, which indicated that the capillary pressure gradually increased. The capillary pressure was about −6 kPa at 46%, 12 kPa at 52% and 80 kPa at 62%. The results showed that the increase of fiber volume fraction made the permeability of the fiber bundle decrease. The resistance of resin in the fiber bundle increased, and the capillary pressure increased.

#### 3.2.4. Influence of Capillary Number on Capillary Pressure and Flow Balance of Infiltration Flow

For fiber cloth of the same kind and fiber volume fraction, there was an optimal capillary number so that the capillary pressure was 0, thus making the flow front flush. The influence of capillary number on capillary pressure when the fiber volume fraction was 46% is shown in Figure 12d. With the decrease of capillary number, capillary pressure decreased gradually. When the capillary pressure was 0, it can be seen from Figure 12d that the capillary number was about 0.026. The experimental data were in good agreement with Tanner’s law, and at the same time, they were very close to the simulation results.

Figure 13 shows the influence of capillary number on the equilibrium of dual-scale infiltration flow. As shown in Figure 13a–d, it is clear that the resin flow rate in the inter-bundle channel was faster, while the resin flow rate in the fiber bundle was slower. At this time, the resin flowing forward would be guided and infiltrated into the fiber bundle by the braided wire, so as to wrap the air inside the fiber bundle and form pores. Due to the large capillary pressure in the fiber bundle, the flow velocity in the two regions was quite different. The larger the capillary number was, the more obvious this phenomenon was, and the larger the size and number of pores that were formed. When the capillary number was small, as shown in Figure 13e,f, there was little difference between the flow velocity of the bundle channel and the fiber bundle, and the flow front was relatively flat with almost no pores. When the capillary number was 0.018, it can be seen from the shape of the flow front that the flow velocity in the fiber bundle was gradually faster than that in the inter-bundle channel. According to the shape of the flow front, there is an optimal capillary number that makes the capillary pressure generated by the fiber zero. In this way, the flow front is uniform, and the porosity can be minimized.

### 3.3. Properties of the Composites

#### 3.3.1. Pore Distribution

Figure 14a,b are cross-sectional microscopic observation results of sample regions 1 and 2, respectively. It can be seen from the figure that the gap filling effect between resin and filament was very good inside the fiber bundle, which means that the inside of the fiber bundle was fully infiltrated. For region 1, macroscopic pores with diameters of about tens to hundreds of microns could be clearly observed at the channels between fiber bundles. These pores were caused by the air wrapping formed by the resin in the channel between bundles. For region 2, microscopic pores with a diameter of about tens of microns could be observed at the channel between fiber bundles, and the size of pores was much smaller than when the capillary number was 0.007. This shows that the channels between fiber bundles were still not fully infiltrated by resin, but the degree of pores formed by air encapsulation was much reduced. The microscopic observation results of the cross-section of the sample obtained in region 3 are shown in Figure 14c. It can be seen from the figure that the gap filling effect between resin and fiber filaments was very good at the channel between bundles, and almost no pores were observed. This means that the channels between fiber bundles were fully infiltrated. In the fiber bundle, microscopic pores with a diameter of several tens can be observed clearly. This shows that the interior of the fiber bundle was not fully infiltrated, and these pores were caused by the air pockets formed in the fiber bundle.

#### 3.3.2. Interlaminar Shear Strength

When the load reached the maximum load, the composite test piece had obvious interlaminar shear failure, as shown in Figure 15a. The interlaminar shear strength of thermoplastic composites is directly related to the pore system between fiber bundles and inside fiber bundles. Through numerical simulation and experimental verification, it is known that capillary number is the main factor affecting the formation of resin infiltration flow pores. Therefore, it is necessary to study the influence of capillary number on interlaminar shear strength of thermoplastic composites. According to Equation (9), the interlaminar shear strength of samples with different capillary numbers is calculated according to the average maximum load, and the interlaminar shear strength of samples with different capillary numbers is shown in Figure 15b. The interlaminar shear strength was 30.5 MPa when the capillary number was 0.007, 32.9 MPa when the capillary number was 0.018 and 30.7 MPa when the capillary number was 0.037. With the increase of capillary number, the interlaminar shear strength increased gradually and then decreased. Moreover, the results show that the capillary number had an obvious influence on the interlaminar shear strength of thermoplastic composites. From the trend of interlaminar shear strength test, it can be seen that there was an optimal capillary number, which made the double-scale infiltration process achieve the best effect. Therefore, the interlaminar shear strength of the composite material was the highest. The optimal capillary number obtained by dual-scale infiltration experiment was 0.026 and the optimal capillary number obtained by simulation research was 0.022. The above experimental results are basically consistent with the test results of interlaminar shear strength.

## 4. Conclusions

In this paper, a general and effective numerical approach was proposed to simulate the balancing of dual-scale infiltration flow in in situ injection molding of continuous fiber reinforced thermoplastic (CFRT) composites. Combined with simulation and experimental research, the effects of injection rate, resin viscosity, fiber volume fraction and capillary number on the dual-scale infiltration flow were studied. Moreover, based on theoretical research, an in situ polymerization injection molding device was developed. Then, continuous glass fiber reinforced thermoplastic pCBT matrix composites were successfully prepared, and the influence of capillary number on pore distribution and interlaminar shear strength of thermoplastic composites was analyzed. This study provides an effective prediction method for dual-scale infiltration flow control in in situ injection molding and provides theoretical guidance for the preparation of high-performance thermoplastic composites.

Key findings of this study include the following:The analysis of numerical simulation results shows that the injection rate and viscosity of resin have significant influence on the infiltration flow. With the increase of fiber volume fraction and capillary number, both the capillary pressure and the flow resistance in the fiber bundle increases, and the flow resistance in the fiber bundle increases. The flow velocity in the channel between fiber bundles is faster, which results in the flow front being unable to flow forward flush.Through the in situ injection molding dual-scale infiltration experiment, the influence of process parameters on capillary pressure and flow balance in dual-scale infiltration flow were obtained. The results show that the numerical model can accurately reflect the unbalanced osmotic flow between and within fiber bundles caused by the changes of injection rate, resin viscosity, fiber volume fraction and capillary number.The infiltration flow velocity in inter- and intra-fiber bundles can be efficiently tuned by the capillary number, which is mainly determined by the injection rate for a specified resin system.Pore distribution and interlaminar shear strength of continuous glass fiber reinforced thermoplastic pCBT matrix composites are affected by capillary number. When the capillary number continues to increase, the pores in fiber bundles gradually increase, while the pores in the channels between fiber bundles gradually decrease. Then the interlaminar shear strength increases gradually and then decreases.

## Figures and Tables

**Figure 1 polymers-13-02689-f001:**
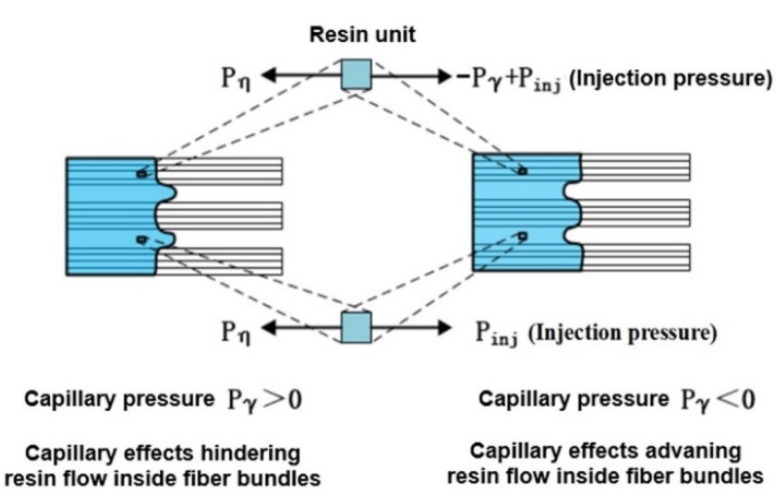
Formation mechanism of dual-scale flow in resin infiltration process.

**Figure 2 polymers-13-02689-f002:**
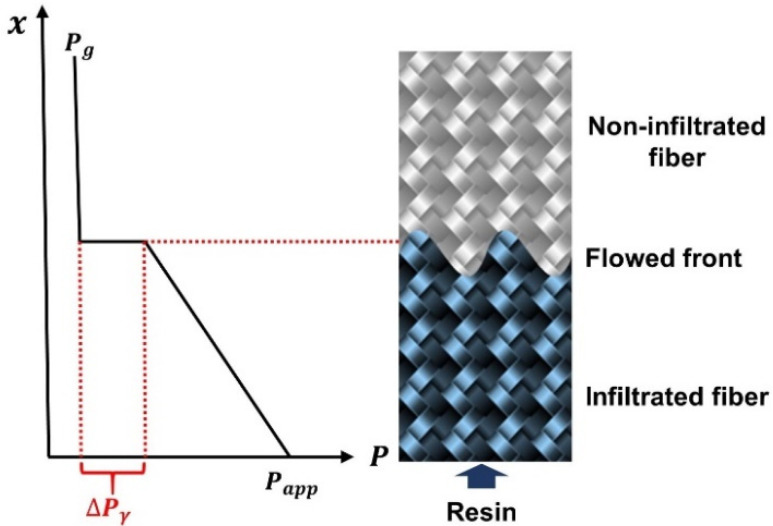
Schematic diagram of resin infiltrated fiber cloth.

**Figure 3 polymers-13-02689-f003:**
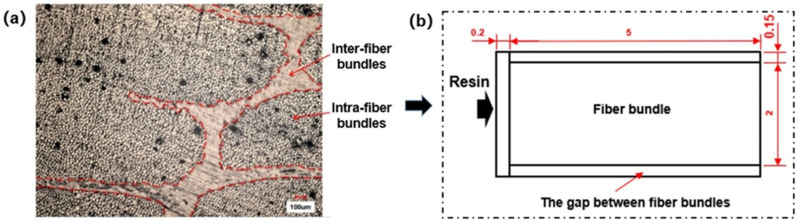
(**a**) Cross section morphology of the fiber reinforced composites under optical microscope. (**b**) The corresponding physical model of fiber reinforcement dual-scale structure established with geometrical unit of mm.

**Figure 4 polymers-13-02689-f004:**
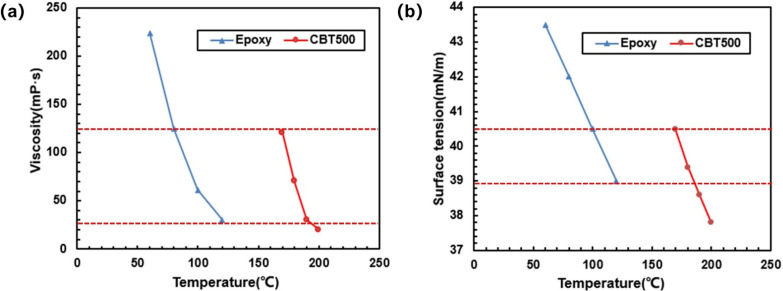
(**a**) The viscosity of CBT resin system and epoxy resin system. (**b**) Surface tension of CBT resin system and epoxy resin system.

**Figure 5 polymers-13-02689-f005:**
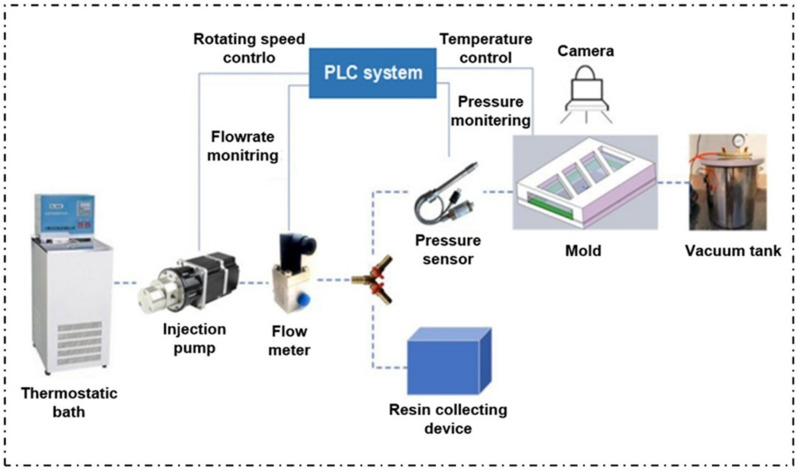
Experimental platform for dual-scale infiltration of in situ injection molding.

**Figure 6 polymers-13-02689-f006:**
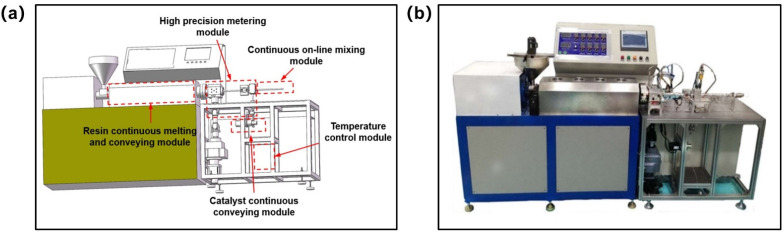
(**a**) Schematic diagram of in situ polymerization injection molding device. (**b**) Physical drawing of in situ polymerization injection molding device.

**Figure 7 polymers-13-02689-f007:**
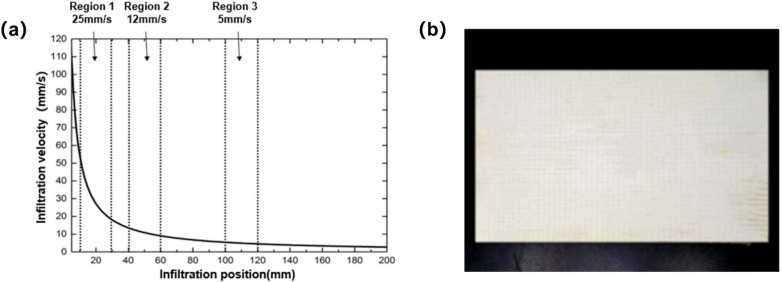
(**a**) Change of infiltration velocity under injection pressure of 20 kPa. (**b**) Continuous glass fiber reinforced thermoplastic pCBT matrix composites.

**Figure 8 polymers-13-02689-f008:**
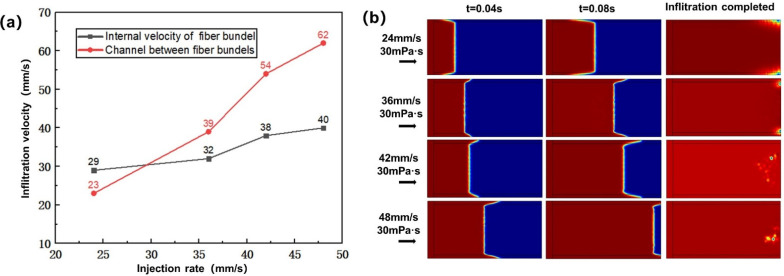
(**a**) Influence of injection rate on infiltration flow velocity. (**b**) Influence of injection rate on infiltration flow front.

**Figure 9 polymers-13-02689-f009:**
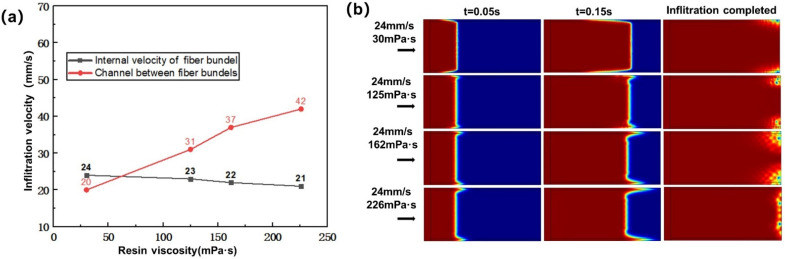
(**a**) Influence of resin viscosity on flow velocity. (**b**) Influence of resin viscosity on flow front.

**Figure 10 polymers-13-02689-f010:**
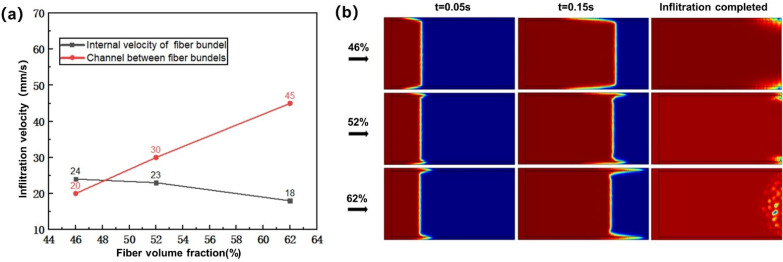
(**a**) Influence of fiber volume fraction on flow velocity. (**b**) Influence of fiber volume fraction on flow front.

**Figure 11 polymers-13-02689-f011:**
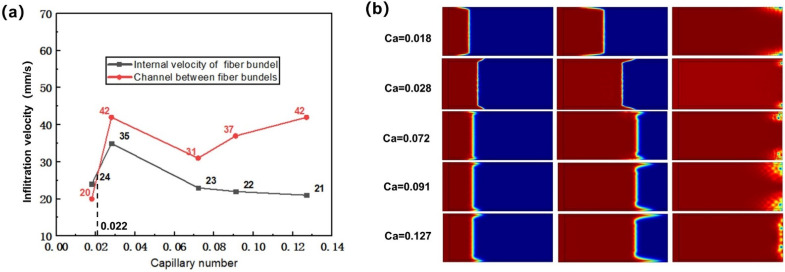
(**a**) Influence of capillary number on flow velocity. (**b**) Influence of capillary number on flow front.

**Figure 12 polymers-13-02689-f012:**
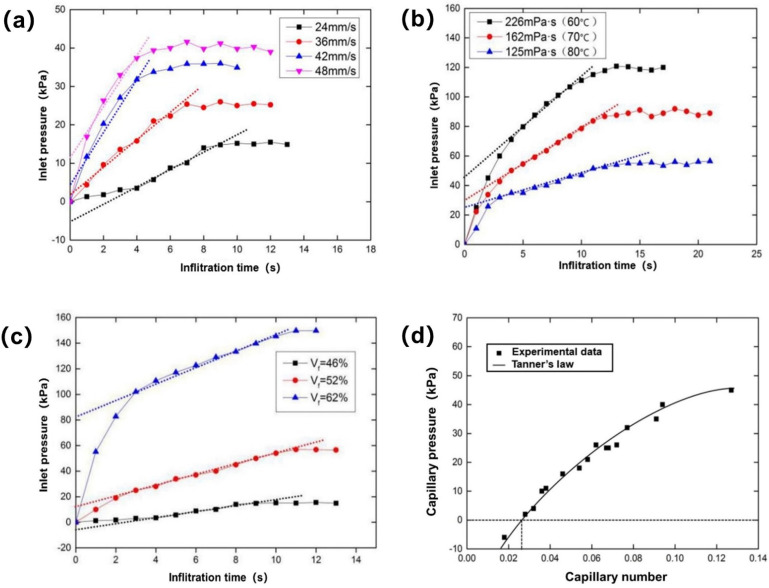
(**a**) Influence of injection rate on inlet pressure variation. (**b**) Influence of resin viscosity on inlet pressure change. (**c**) Influence of fiber volume fraction on inlet pressure. (**d**) Influence of capillary number on capillary pressure during infiltration.

**Figure 13 polymers-13-02689-f013:**
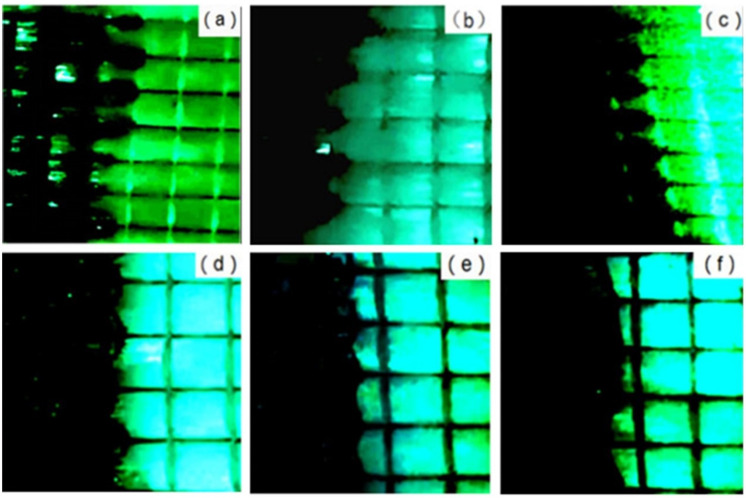
The influence of capillary number on flow front morphology: (**a**) 0.094; (**b**) 0.072; (**c**) 0.058; (**d**) 0.038; (**e**) 0.028; (f) 0.018.

**Figure 14 polymers-13-02689-f014:**
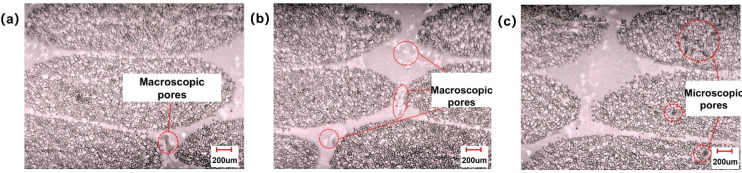
Microscopic morphology of pore distribution under different capillary numbers: (**a**) 0.007; (**b**) 0.018; (**c**) 0.037.

**Figure 15 polymers-13-02689-f015:**
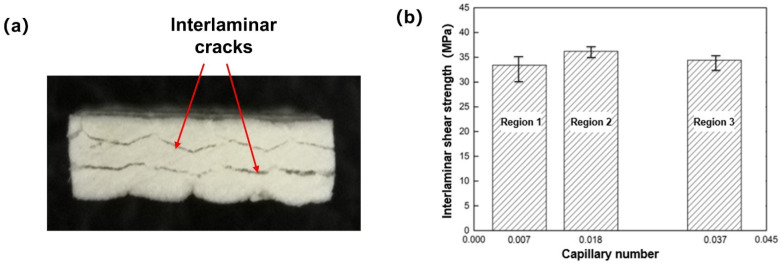
(**a**) Delaminated samples after ILSS test. (**b**) Influence of capillary number on interlaminar shear strength.

**Table 1 polymers-13-02689-t001:** Main technical parameters of in situ injection molding dual-scale infiltration experimental platform.

Parameter	Value
Heating temperature of mold (°C)	≤300
Injection flow rate (mL/min)	0–1000
Injection pressure (MPa)	≤1
Measurement accuracy (mL/min)	±1
Pressure accuracy (MPa)	±0.001
Resin temperature control (°C)	±1

**Table 2 polymers-13-02689-t002:** Main technical parameters of in situ polymerization injection molding equipment.

Parameter	Value
Heating temperature of mold (°C)	≤300
Resin flow rate (mL/min)	100–900
Catalyst flow rate (mL/min)	0–100
Injection pressure (MPa)	≤1
Measurement accuracy (mL/min)	±1
Pressure accuracy (MPa)	±0.001
Resin temperature control (°C)	±1
Mixture ratio	0:1–1:1
Mixing mode	SK static mixing head

## Data Availability

Not applicable.

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
