# Peer review of "A General and Efficient Approach for the Dual-Scale Infiltration Flow Balancing in In Situ Injection Molding of Continuous Fiber Reinforced Thermoplastic Composites"

_polymers, 2021, doi:10.3390/polym13162689_

Round 1

Reviewer 1 Report

The submitted article “polymers-1334244-v1” entitled: “A General and Efficient Approach for the Dual-Scale Infiltration Flow Balancing in In-Situ Injection Molding of Fiber Reinforced Thermoplastic composites” presents an interesting numerical approach to simulate the balancing of dual-scale infiltration flow in in-situ injection molding of thermoplastic composites. Stokes and Brinkman equations are used to describe the infiltration flow in inter- and intra-fiber bundles. Experimental results are also used to validate the proposed numerical model. The paper falls within the scope of the Journal. The manuscript is well-structured and well-written. Figures are helpful. The following minor suggestions could contribute to the final improvement of the paper:

  1. Although the presented research is of high theoretical and practical meaning, its practical side is not well emphasized. Discussion to conclusions summarizing the obtained results in the light of practical applications is strongly recommended.

  1. Discussion and further commentary of Figure 1 would be helpful to comprehend the formation mechanism of dual-scale flow in resin infiltration process. Further, Figure 5 seems rather trivial, unless some explanations or notes in the photograph of the polymerization injection molding device are added.

  1. Parameters affecting the interlaminar shear strength could be commented and further explained.

Reviewer 2 Report

Please use the journal template for the manuscript.

Please explain the abbreviation CBT. 

"Because CBT resin will solidify at low temperature, and the melting temperature of CBT resin is 200℃. After the experiment, the cleaning
process is very complicated, so epoxy resin is proposed to replace CBT resin in the dual-scale infiltration experiment. Because the optimal capillary
number does not change when different resins soak the same volume fraction fiber cloth, the experimental results are also applicable to CBT resin." - so Authors are indicating that the results for any epoxy resin are viable also for other types of resins? Sounds doubtful.

In section 2.4.1.1. Authors are writing that epoxy resin replaced CBT, but then Authors are writing about preparation of CBT matrix composites, strange. Please specify.

I believe that the heating tempearture of mold during tests should be specified more precisely than <300 degrees.

Authors are mentioning in the paper the raw materials not specified before. Please correct.

"The resin viscosity was set at 226,162,125 and 30 mPa·s" - how Authors changed the resin viscosity?

"the fiber volume fraction is 46, 52 and 62%. " - why such strange loadings of fiber?

"Figure 17(a) and Figure 17(b) are cross-sectional microscopic observation" - how was the cross-section prepared?

"For region 1" - what region?

Figure 17 - please enhance the scale, it is not visible now.

Figure 19 - please enhance the quality.

Generally it is very strange that the Authors are claiming that they chosen some epoxy resin and they are comparing the results to other type of resin. Why CBT, why not a different type?

Reviewer 3 Report

The manuscript presents very good work related water purification and going to be interesting for the readers.

Some minor comments are as follows.

  1. Authors need to include some interesting data in the abstract part of the manuscript.
  2. English must be improved.
  3. Novelty of the work be established.
  4. All the important results reported be compared in a tabular form to establish the superiority of the work.
  5. What is the significance and outcome of this study?
  6. Avoid the typo errors and unnecessary space in the written manuscript.

  1. Authors need to add future prospective of presented research in the conclusion part of the manuscript.
  2. Authors must need to incorporate  recent reference related to the topic of the manuscript to make it more interesting for the readers.

Round 2

Reviewer 2 Report

Good job with the review, everything in order after corrections.